# Ablation provides key macronutrients (nitrogen and phosphorous) to glacier ice algae in NW Greenland

B. Gill-Olivas ®[1] ✉, P. Forjanes[2,3], T. C. Turpin-Jelfs ®[1], A. M. Anesio ®[1], L. G. Benning ®[2,4] & M. Tranter ®[1]

Dissolved phosphorus (P) is thought to limit the growth of glacier ice algae, which darken the surface of the Greenland Ice Sheet (GrIS) and enhance surface melt. This contention is largely based on the low-level of P concentrations, which are often below the limit of detection of conventional methods. Here, we propose that low-level nutrient analysis is essential to understand the macronutrient limitation of glacier algal growth on the GrIS. We sample surface (≤5 cm) and shallow (5–10 and ~80–90 cm depth) ice from two sites in NW Greenland and measure the nano-molar concentrations of dissolved nitrogen (N) and P using a custom-built continuous segmented flow analyser. Quantifiable concentrations of these macronutrients are present, along with glacier ice algal abundances comparable to those of the Dark Zone, a zone of biologically enhanced ice sheet melting. Mass balance calculations for each site indicate that N and P released during seasonal ablation exceed the amounts incorporated into glacier ice algal biomass. These results suggest that dissolved macronutrients are unlikely to limit glacier ice algal growth on the Greenland Ice Sheet.

The melt zones on glaciers and ice sheets worldwide are hotspots for microbial activity during the summer[1–3]. Surface ice on the Greenland Ice Sheet (GrIS) has consistently darkened during the summers of the last 20 years[4–6] due to the growth of Streptophyte glacier ice algae[7,8], a microalgae with a characteristically dark purple phenolic pigment[9–11]. The growth of these algae decreases the surface albedo by between 0.13 and 0.25 in areas with low ($10^3$ cells mL$^{-1}$) and high ($10^4$ cells mL$^{-1}$) biomass, respectively, increasing surface melt rates[12]. This seasonal darkening is most noticeable in the Dark Zone, identified as the area along the South-West margin of the ice sheet[6]. Glacier ice algae are endemic to most ice surfaces[3], yet the controls or promoters of their growth are not well understood.

One of the likely controls on growth is macronutrient limitation. Carbon (C), nitrogen (N) and phosphorous (P) are the building blocks of life, being essential for cellular mass and metabolic activity[13]. Carbon

is readily available to primary producers at the ice surface from atmospheric $CO_2$ fixation. However, atmospheric N and P inputs to the ice surface are thought to be limited, other than those derived indirectly from snow and ice melt. More recently, attention has focused on the potential of atmospheric dust to fertilise the ice surface[14–16]. Bioavailable dissolved inorganic N (DIN; mostly ammonium [$NH_4^+$] and nitrate [$NO_3^-$]) is primarily derived from snow and ice melt[17,18], or from nitrogen fixation in cryoconite holes[19]. Bioavailable P, often quantified as soluble reactive phosphorous (SRP), is largely derived from the chemical weathering or bio-mining of mineral dusts containing P-bearing minerals such as apatite, originating either from wet (snow and more often now rain) or dry deposition or melted out of meteoric ice[14,15,20,21].

A few studies assessed macronutrient concentrations in surface ice and meltwater samples collected within the Dark Zone of the GrIS,

[1]Department of Environmental Science, Aarhus University, Roskilde, Denmark. [2]Interface Geochemistry, GFZ Helmholtz Center for Geosciences, Potsdam, Germany. [3]Departamento de Mineralogía y Petrología, Universidad Complutense de Madrid, Madrid, Spain. [4]Department of Earth Sciences, Freie Universität Berlin, Berlin, Germany. ✉ e-mail: b.gillolivas@envs.au.dk

but conclusions may be limited by the relatively high detection limits of the analytical methodology in relation to the low concentrations of dissolved N and P species[18,22]. Dissolved N and P concentrations in surface ice melt are typically low[17,18,23], often sub-micromolar, and near the detection limit of most standard automated spectrophotometric and IC analytical methods previously used in these environments. Recent studies suggest that concentrations of various nutrients may be higher in the micromelt (i.e., a thin layer of melt water of the surface of melting ice crystals) than in the bulk ice crystals[24]. However, even in these samples, the concentration of $PO_4^{3-}$, $NO_3^-$, and $NO_2^-$ was largely below the limit of detection (LOD)[24]. Dissolved organic N and P (DON and DOP) may be present in much higher concentrations[22], but it is currently unclear whether these species are readily utilised by the glacier ice algae.

We contend that our knowledge of the potential limitation of the macronutrients on glacier algal growth is hampered by the high limit of detection of most conventional spectrophotometric analytical methods. We illustrate the need to accurately measure low concentrations by determining the mass of this macronutrient released from melting ice during the ablation season and comparing this with the mass of macronutrient contained in the pool of algae in the surface ice. We show that nutrients may appear to be limiting due to their low, nanomolar concentrations, but that since only a very small mass of nutrient is needed to maintain the population of glacier ice algae at the surface, these low concentrations, otherwise thought to denote oligotrophic conditions, may be sufficient to meet algal metabolic demands.

Here, we analysed samples from two sites in northwest Greenland (Fig. 1b), where seasonal darkening has been observed[25–27], one on the Qaanaaq Ice cap, and the other on the nearby GrIS, and quantified the concentration of macronutrients present in the surface ice. We collected shallow ice cores (-80–90 cm in depth) and analysed three primary sections (Fig. 1c): (1) the topmost 5 cm of ice – the weathering crust (WC); (2) the transition zone (TZ) linking the weathering crust to the underlying unweathered ice (5–10 cm depth); and 3) the bottom 5 cm from the base of each ice core – representing the fully unweathered (UW) ice (density > 0. 91 g cm$^{-3}$[28]). In addition, we collected adjacent surface WC samples from a -2 m$^2$ area using a field-conditioned ice axe to scrape off the top layer (-3 cm) of the weathered ice. The loose surface ice samples were homogenized and stored in 2 L

Whirlpak® bags. Both ice core samples and surface samples were melted in a field laboratory and filtered using inline 0.45 μm PES filters into acid washed falcon tubes and frozen at −20 °C until analysis. We used a custom-built continuous segmented flow analyser fitted with a 250 cm flow path[29]. The 10- to 100- fold increase in pathlength, compared to most commercially available nutrient analysers, enabled accurate quantification of nanomolar concentrations of SRP and DIN. Alkali persulphate digestions[30] were undertaken to quantify DOP and DON concentrations in our samples. We then used macronutrient mass balance calculations to determine how much meteoric ice melt is required to provide the macronutrients we estimate are held within the surface glacier ice algae, and then examined stochiometric ratios of N:P and assessed the potential for N and P limitation. Finally, we make the case that low-level nutrient analyses are an essential prerequisite for conducting nutrient limitation studies on the GrIS.

## Results and Discussion
### Nutrient availability at and below the weathering crust
Our custom-built continuous segmented flow analyser was able to quantify SRP and DOP in all samples collected. Median SRP concentrations were broadly similar in both the Qaanaaq ice cap and Greenland ice sheet samples (Fig. 2a). Values ranged from $35 ± 12$ nM and $28 ± 8.0$ nM for surface (WC) samples and $18 ± 6.0$ nM and $20 ± 7.0$ nM for the UW samples at Qaanaaq and GrIS, respectively. Median DOP, defined as total dissolved P (TDP) minus SRP, was lower at the GrIS site than in Qaanaaq, particularly at the surface (WC; $47 ± 19$ nM and $96 ± 49$ nM, respectively). The transition zone (TZ) samples from Qaanaaq showed wider ranges of concentrations than the GrIS site. The median SRP and DOP concentrations at the surface (WC samples) were higher than the sub-surface concentrations (TZ and UW samples) at both sampling locations. However, the differences were only statistically significant for DOP at Qaanaaq (Supplementary Information, Figure S2). We caution that the DOP concentrations measured at the ice surface may be overestimated, since lysis of algal cells may occur during the melting of the samples and/or during filtration. We estimate that lysis of only between 2.8 to 3.8% of the cells counted at Qaanaaq (mean value: $2.4 × 10^5$ cells mL$^{-1}$) and GrIS sites (mean: $0.75 × 10^5$ cells mL$^{-1}$), respectively, would be sufficient to produce the DOP concentrations measured in the ice surface samples (see Supplementary Information text and, Table S1).

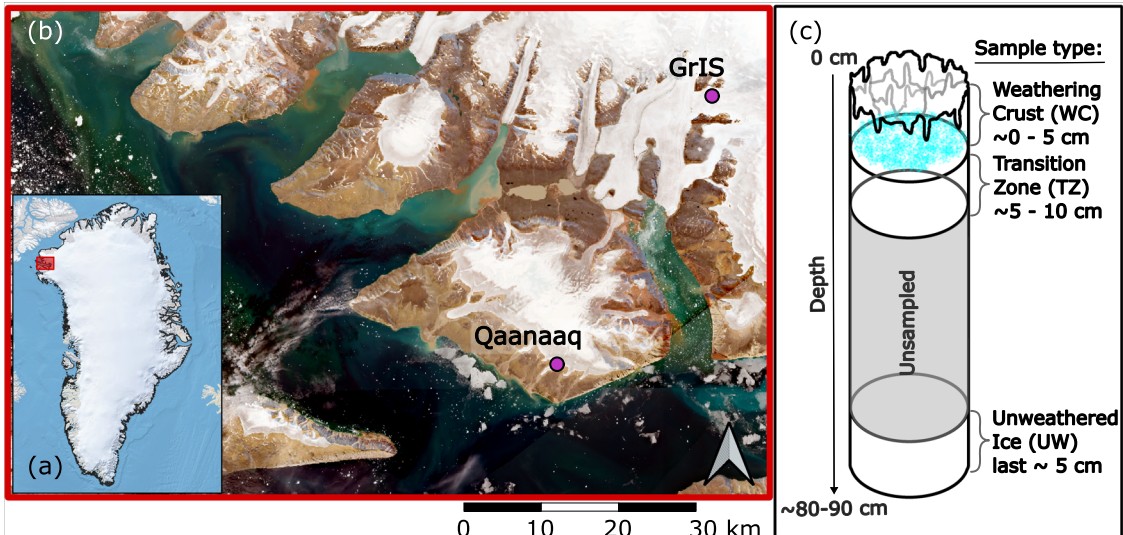

**Fig. 1 | Sampling location and samples collected. a** Basemap of Greenland with a red box indicating sampling site location. **b** Sentinel 2 L2A true colour satellite image of sampling site on 27/07/2023 (accessed through Copernicus on 14/08/2025). Purple circles indicate the Qaanaaq (approximate sampling location: N 77° 30.092', W 69° 09.691') and GrIS (approximate sampling location: N 77° 48.259', W 68° 21.083') sampling sites. **c** Schematic diagram of a shallow ice core and the three depths collected when taking ice core samples: weathering crust (WC), transition zone (TZ), and unweathered ice (UW).

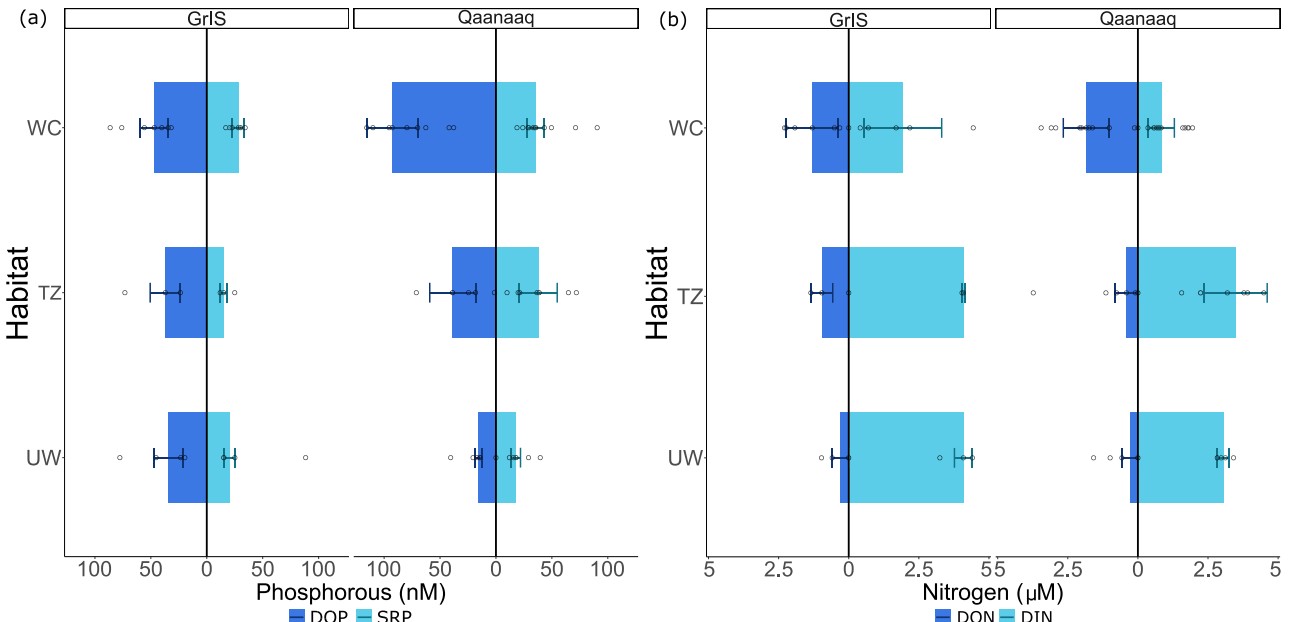

**Fig. 2 | Phosphorous and nitrogen concentrations across sampled habitats.** **a** Median concentration of dissolved organic phosphorous (DOP) and soluble reactive phosphorous (SRP) for each sample type. **b** Median concentration of dissolved organic nitrogen (DON) and dissolved inorganic nitrogen (DIN) for each sample type. Sample types shown are weathering crust (WC), transition zone (TZ) and unweathered ice (UW). Error bars represent the median absolute deviation (MAD) for these values and hollow points show individual measurements. Organic fractions are seen in cornflower blue and the inorganic fraction is seen in light blue.

**Table 1 | Median values and median absolute deviation (MAD) of Total Dissolved Phosphorous (TDP), Soluble Reactive Phosphorous (SRP) and Dissolved Organic Phosphorous (DOP), Total Dissolved Nitrogen (TDN), Dissolved Inorganic Nitrogen (DIN), Dissolved Organic Nitrogen (DON), Nitrate (NO$_3^-$), Nitrite (NO$_2^-$) and Ammonium (NH$_4^+$) concentrations in weathering crust (WC), transition zone (TZ) and unweathered ice (UW) samples collected at GrIS and Qaanaaq**

| | WC | | | | TZ | | | | UW | | | |
|---|---|---|---|---|---|---|---|---|---|---|---|---|
| | GrIS | | Qaanaaq | | GrIS | | Qaanaaq | | GrIS | | Qaanaaq | |
| | Median | MAD | Median | MAD | Median | MAD | Median | MAD | Median | MAD | Median | MAD |
| TDP (nM) | 71 | 19 | 142 | 71 | 49 | 0 | 75 | 41 | 55 | 26 | 36 | 12 |
| SRP (nM) | 28 | 8 | 35 | 12 | 15 | 4.7 | 38 | 25 | 20 | 7 | 18 | 6 |
| DOP (nM) | 47 | 19 | 96 | 49 | 37 | 20 | 39 | 39 | 34 | 19 | 16 | 4.6 |
| TDN (µm) | 3.4 | 2.8 | 2.8 | 1.3 | 5 | 2.5 | 3.8 | 1 | 4 | 1.6 | 3.2 | 1.3 |
| DIN (µm) | 2.2 | 2.6 | 0.8 | 0.7 | 4.1 | 0.1 | 3.5 | 1.7 | 4.3 | 0.9 | 3 | 0.3 |
| DON (µm) | 1.3 | 1.4 | 2 | 1.5 | 1 | 0.6 | 0.6 | 0.8 | 0.3 | 0.4 | 0.3 | 0.4 |
| NO$_3^-$ (µm) | 0.7 | 1 | 0.1 | 0.2 | 1.4 | 0.6 | 0.9 | 0.3 | 1.7 | 0.7 | 1.3 | 0.1 |
| NO$_2^-$ (nm) | 14 | 5.4 | 25 | 6.9 | 3.1 | 2.8 | 5.5 | 6.6 | 2 | 2.6 | 3.7 | 2.6 |
| NH$_4^+$ (µm) | 1.7 | 1.9 | 0.8 | 0.6 | 2.6 | 0.6 | 2.6 | 1.0 | 2.7 | 0.7 | 1.8 | 0.4 |

Total P (TP) concentrations measured in ice cores from the North Greenland Eemian (NEEM) drilling site were high and variable (7 – 637 nM), but it is important to note that TP was analysed on unfiltered samples, and so likely contain mineral bound-P. These TP concentrations are dependent on the age and climate interval that was sampled, and correlate with the dust content[20], suggesting that particulates in the unweathered ice have the potential to be a source of at least some dissolved P. We have determined TDP on filtered samples and so have excluded particulate P contributions. However, our TDP values (means ranging from 36–142 nM; Table 1) lie within the wide range of TP values measured at NEEM. The SRP concentrations reported here (Table 1), are an order of magnitude higher than those measured in firn cores from the North East Greenland Ice Stream (NEGIS) ice core sampling site, within the GrIS accumulation zone (mean SRP at NEGIS: 2.7 nM[21]). The concentration of SRP measured in our samples were mostly within the range measured in NEEM ice core samples (SRP within NEEM:

3.0–32 nM[20]), however, some of the SRP concentrations in our samples were up to four times higher than the upper values found at NEEM (Table 1).

Dissolved N-species were predominantly inorganic in all samples (Fig. 2b). DIN concentrations were lowest at the surface. The median surface DIN concentration was $0.8 \pm 0.7 \, \mu M$ and $2.2 \pm 2.7 \, \mu M$ at Qaanaaq and GrIS, respectively (Table 1). Whilst in UW samples the concentrations were and $3.0 \pm 0.3 \, \mu M$ and $4.3 \pm 0.9 \, \mu M$, at Qaanaaq and GrIS respectively. The dominant DIN species was NH$_4^+$, which comprised roughly 60% of the total DIN in the GrIS samples (Fig. 3), and up to 85% in surface ice at Qaanaaq (Fig. 3). The scatter in the dataset limits the ability to detect statistical differences between depths and sites; statistically significant differences are presented in the supplementary information (Figure S3–S6). Our custom-built analyser also allowed us to quantify very low nitrite (NO$_2^-$) concentrations (LOD = 1.5 nM). Although NO$_2$ only makes a small fraction of the DIN

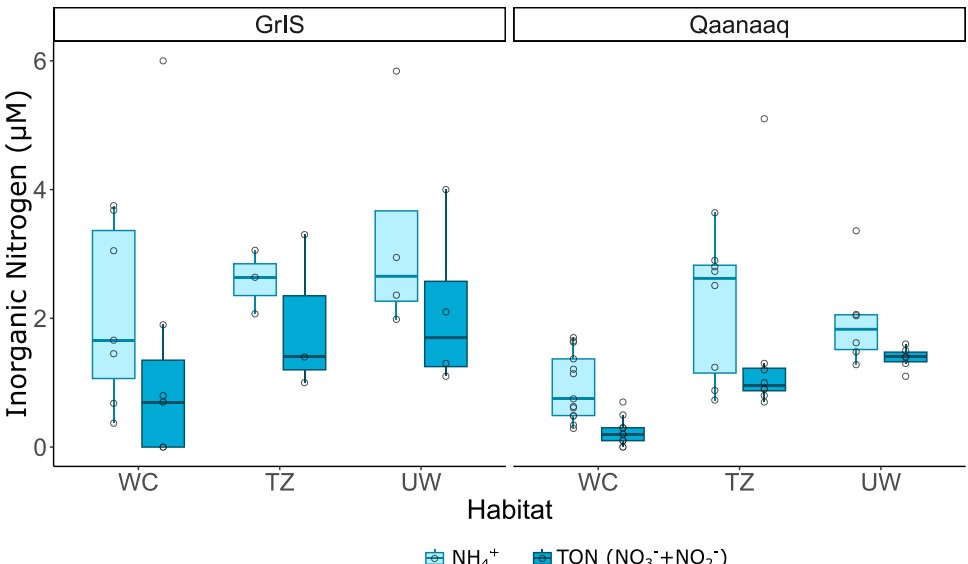

**Fig. 3 | Nitrogen speciation across sampled habitats.** Boxplots showing concentrations of inorganic nitrogen species (ammonium [NH₄⁺], in light blue, and total oxidized nitrogen [TON], in darker blue) across samples: weathering crust (WC), transition zone (TZ), and unweathered ice (UW), in GrIS and Qaanaaq. Boxes indicate median and interquartile ranges, while hollow points show individual measurements.

pool, our data show that the concentrations were between five and 10 times higher in surface samples ($14 \pm 5.4$ nM and $25 \pm 6.9$ nM at GrIS and Qaanaaq, respectively) in comparison with those at depth (between $2.0 \pm 2.6$ nM and $3.7 \pm 2.6$ nM in UW samples). DIN concentrations were lower than those measured during the early melt season in wet surface ice and snow within the Dark Zone ($2.3$–$5.2$ μM[18]), when there were low light levels at the ice surface due to the overlying wet snow. However, the median concentrations in unweathered ice samples were similar ($2.8$–$6.2$ μM[18]). Other studies, conducted near Kangerlussuaq (Southern Greenland), later in the melt season, showed similar surface $NO_3^-$ concentrations to ours (mean $0.6$ μM[31]). Hence, the timing of our sample collection is likely linked to the relatively low surface concentrations of DIN and $NO_3^-$. The relative increase in $NO_2^-$ at the surface (Table 1) could arise from atmospheric deposition or, potentially, be indicative of nitrification processes, which are associated with cryoconite granules[32,33].

Dissolved organic N concentrations were highest at the surface (median concentrations of $2.0 \pm 1.5$ μM and $1.3 \pm 1.4$ μM at Qaanaaq and GrIS, respectively; Table 1), with lowest concentrations at the base of the ice cores (median UW concentrations of $0.3$ μM at both Qaanaaq and GrIS; Table 1). As with DOP, the higher DON surface concentrations at Qaanaaq might be due to a methodological artefact arising from cell lysis (see Supplementary Information text and Table S1), in this case, lysis of roughly 2.2% or 3.5% of the cells present would produce similar concentrations of DON.

The general pattern observed at both sites is that DIN is depleted at the surface relative to the deeper unweathered ice. This is consistent with the active uptake of nutrients by microbes, including glacier ice algae, that bloom primarily at the ice surface (WC samples). This matches a recent study, which suggests that glacier ice algae can utilise both $NO_3^-$ and $NH_4^+$ as sources of N at similar rates[34]. Elevated concentrations of DON and DOP at the surface may be, in part, an artefact of sample processing, but could also arise as a result of algae accumulation and exudation of organic compounds under natural conditions. The concentrations of DON at these sites were relatively low compared to previously reported DON concentrations measured on the surface of the GrIS during spring, ~1300 km to the south, on the GrIS in the Dark zone site near Kangerlussuaq ($5.1$ to $14.0$ μM[22]), despite between seven to 10 times higher cell counts found at our sites. Although methodological artefacts may have also inflated the DON

concentrations measured in previous studies, it is also possible that natural, overwinter lysis of dormant algal cells may contributed to the high concentrations found in spring[22]. Natural lysis of cells occurs due to factors such as viral[35] or fungal infection[36], and studies have found approximately 10% of cells within a population have been found to be inactive or dead[34], which may impact the concentrations of organic N and P measured.

The exception to this pattern is SRP, which is elevated in the surface ice and may arise from surface dust dissolution and mineralization processes. Although we have only a few measurements of particle loadings, these show higher values in the samples from Qaanaaq (average of $26$ mg L$^{-1}$; $n = 2$) than at the GrIS site (average of $8$ mg L$^{-1}$; $n = 3$). The masses include both organic and inorganic particles, yet the higher particle loading is consistent with enhanced dry deposition of local dust at Qaanaaq versus the more remote GrIS site, which is further inland and at a higher elevation. The surface particles in the Qaanaaq samples consisted of approximately 95% wt. % quartz, with minor amounts of feldspar, mostly K-feldspars, and illite. This is consistent with the local geology near Qaanaaq, which consists primarily of quartz sandstones from the Thule sedimentary supergroup[37]. The dust collected from the GrIS site, while also dominated by quartz, contained amphiboles and a higher proportion of feldspars (including plagioclase feldspars), matching the local geology[38]. Although hydroxyapatite was found neither in the Qaanaaq nor in the GrIS samples according to XRD and SEM analyses, some of the minerals quantified in our dust samples (in particular the feldspars) may contain trace concentrations of P and so could provide some of the P to the surface ice[39]. This differs from the mineral composition of the ice-surface dust quantified in the dark zone near Kangerlussuaq[14], where a link between hydroxyapatite in the dust and algal biomass could be established. In the samples from this study, the identified K-feldspars can also trace amounts of $NH_4^{+}$[40,41], which could also supply some N to the system. Regardless of the potential contributions from local dust, glacier ice algae have been documented to contain P granules[34], indicating that they are able to store P through luxury P uptake mechanisms, as seen in other microalgae[42]. This P can then be utilised during periods when there would otherwise be P starvation. Finally, DOP is often considered more recalcitrant than SRP, but there is evidence of phosphatase activity by microbes on ice sheet surfaces[43]. These were primarily associated with cryoconite granules and cryoconite holes, which were

also present at both the Qaanaaq and GrIS sites. Together, the concentrations of SRP and DOP measured in the ice surface samples in this current study suggest that the surface ice environments in the Qaanaaq Ice Sheet and the GrIS in this area have sufficient P to sustain the documented glacier ice algal blooms.

Our data show that dissolved inorganic species of both macronutrient N and P are present along with glacier ice algal cell concentrations typical of those found in melting surface ice of the Greenland Ice Sheet[20,31]. This in turn suggests that macronutrients in this site do not limit glacier ice algal growth in the classical sense, where complete absence denotes limitation. In these sites, the macronutrients are present in nanomolar concentrations and so could perhaps limit growth if the rate of their uptake is concentration dependent. However, we show below that the annual release of macronutrient N and P dissolved inorganic species from unweathered ice by melting alone exceeds that of the mass of nutrients estimated in the glacier ice algae at the surface, which may mitigate any kinetic uptake limitation.

## How much algal growth can ice melt support?

Summer melting progressively lowers the ice surface, effectively replacing the pre-existing ice surface with deeper ice. Hence, ice melt (ablation) provides a continual fresh source of nutrients to the surface equivalent to the N and P content of the deeper ice. Here, we calculate how much ablation is required to provide the mass of N and P in average glacier algae cells at our sites (Supplementary Information, Table S1). Based on our measurements, we take the average cell counts for each site ($2.4 \times 10^5$ cells $mL^{-1}$ in Qaanaaq and $0.75 \times 10^5$ cells $mL^{-1}$ in GrIS), and assuming that the average *Ancylonema spp*. cell contains 8.8 pM C $cell^{-1}$[44], we calculated that the mass of C in the glacier ice algal cells is $1.8 \times 10^6$ pmol C $mL^{-1}$ in Qaanaaq and $7.1 \times 10^5$ pmol C $mL^{-1}$ in GrIS. We then used the median intra-cellular molar C:N and C:P ratios of the glacier ice algal cells of 19 and 509, respectively[34], and this resulted in the mass of N and P stored in the glacier ice algae of $9.3 \times 10^4$ pmol N $mL^{-1}$ and $3.5 \times 10^3$ pmol P $mL^{-1}$ at Qaanaaq, and $3.7 \times 10^4$ pmol N $mL^{-1}$ and $1.4 \times 10^3$ pmol P $mL^{-1}$ at GrIS respectively (Supplementary Information, Table S1). Finally, the ablation in units of cm, derived from mL (or cm³) of melt per cm² of ice surface, required to give rise to these algal N and P masses from the DIN and SRP held in deeper meteoric ice was obtained by simply dividing the mass of algal N or P by the average DIN and SRP of the deeper ice (Supplementary Information, Table S1). The volume of water was converted into ablation of ice by dividing by the ice density 0.91 g $cm^{-3}$, assuming that 1 g of unweathered ice melt has a volume of 1 cm³[28]. Roughly 212 cm of ablation provides sufficient SRP to balance the estimated P stored within biomass at the surface of the Qaanaaq ice cap, as does 76 cm of ablation at the GrIS site. These values are lower than estimates of recent annual ablation at similar elevations on the Qaanaaq ice cap (~230 cm $year^{-1}$) and the GrIS site (~100 cm $year^{-1}$)[45]. The ablation required to provide sufficient N from DIN in unweathered ice is smaller, being 34 cm at Qaanaaq and 9.5 cm at the GrIS site. These calculations assume that all of the DIN and SRP is taken up by the glacier ice algae; in reality, the amount of ablation required will be greater because the DIN and SRP uptake efficiency of the algae is unlikely to be 100%, and there is a nutrient demand from other microbes in the microbial ecosystem[46]. Satellite imagery of the area suggests that 2023 was characterised by particularly high cell abundances[47]. Yet, it appears that there is sufficient DIN and SRP in the deeper unweathered ice to provide the mass of N and P in the surface glacier ice algal cells, given the present ablation rates at both sites. Thus, at these sites, there is no need for additional sources of P or N from, for example, mineral dust, as has been suggested to be the case in the Dark zone, near Kangerlussuaq[14].

Clearly, inclusion of DOP and DON in the calculations decreases the amount of ablation that is required. The annual ablation that is required to produce the amount of N and P in the surface glacier algal cells falls to 32 cm and 105 cm at Qaanaaq, and 10 cm and 27 cm at the GrIS site, respectively. This too assumes that the glacier ice algae can directly take up DOP and DON with an efficiency of 100% and/or that heterotrophic microorganisms take up and recycle DOP and DON to inorganic forms with 100% efficiency (i.e., without loss to downstream ecosystems).

## Could nitrogen or phosphorous limit growth in the weathering crust?

Previous studies have used the low concentration of P in supraglacial environments, often below the limit of detection, which often ranges from 0.32 μM[48] to 0.13 μM[18], to argue that P is a limiting factor for microbial growth in these environments[14,48,49]. The detection limit of our custom-built analyser (2 and 5 nM for SRP and TON) allows us to actually determine the concentrations of both N and P at lower levels than in these previous biogeochemical studies. Nutrient uptake in healthy algal cells often follows the overall mean or median stochiometric ratios, but this is also dependent on environmental stresses, such as changing light intensity, nutrient limitation and temperature shifts[50–53]. Furthermore, microalgae may prioritise P uptake and storage after periods of starvation[42]. Therefore, dissolved N:P ratios within environmental samples should be treated as indicative of potential limitation to glacier ice algal growth, and bshould e interpreted with caution. Due to the impact of the dark purple pigment in glacier ice algae on surface albedo[10], nutrient limitation in glacier surface samples has often focused on processes affecting *Ancylonema spp*. growth. Recent imaging of *Ancylonema spp*. cells by scanning electron microscopy (SEM), combined with energy-dispersive X-ray spectroscopy (EDS), determined within individual cells mean intracellular N:P ratios of 26:1, although there was a large range of N:P values (18:1 to 43:1)[34]. *Ancylonema spp*. is part of a wider microbial ecosystem, which includes species with potentially shared or have unique adaptations to different possible limiting factors. The N:P ratios of the bulk organic matter on other sites on the GrIS surface include both an assemblage of microbes and allochthonous matter, and have been reported to have a higher mean N:P ratio of 73:1[44]. Considering that in our samples, both DIN and SRP were always detected in the weathering crust samples, and given that the estimated magnitude of ablation over the summer can provide N and P in excess of the nutrient stored in the glacier ice algae, it is unlikely that these macronutrients limit growth at the sites studied on times scales equivalent to the ablation season (often, up to several months). It is also worth noting that there is a positive correlation between bare ice duration (and consequently melt season length) and surface darkening[54]. This may be partially due to a feedback effect, where increased melting releases more nutrients into the system, further sustaining biological activity. However, the distribution of glacier ice algal cells is very heterogeneous[55], and ablation rates vary greatly throughout the summer at our and many other Arctic sites[56–58], so there are likely to be occasions when nutrient limitation occurs either locally or for periods of days. Examples include localised surface depressions where water-borne transport and deposition have concentrated cells, and prolonged periods of very low ablation rates, in light conditions where ice algae can otherwise grow, but the replenishment of nutrients taken up by the cells via ablation is low.

The ablation required to provide the surface ice algae with N and P is smaller for N than P, suggesting that P will become limiting in these circumstances. It is instructive to examine the dissolved phase N:P ratios to visualise how close the weathering crust is to P limitation. We first examine inorganic N:P ratios, since these are a readily available form of nutrients, where the potential for processing artefacts in our data set is much more limited. Figure 4 a) shows surface (WC) samples have lower inorganic N:P ratios than the deeper (TZ and UW) samples. There is a broad range of surface values at the GrIS site, with a median value of 106:1, while there is a narrower range of surface values at

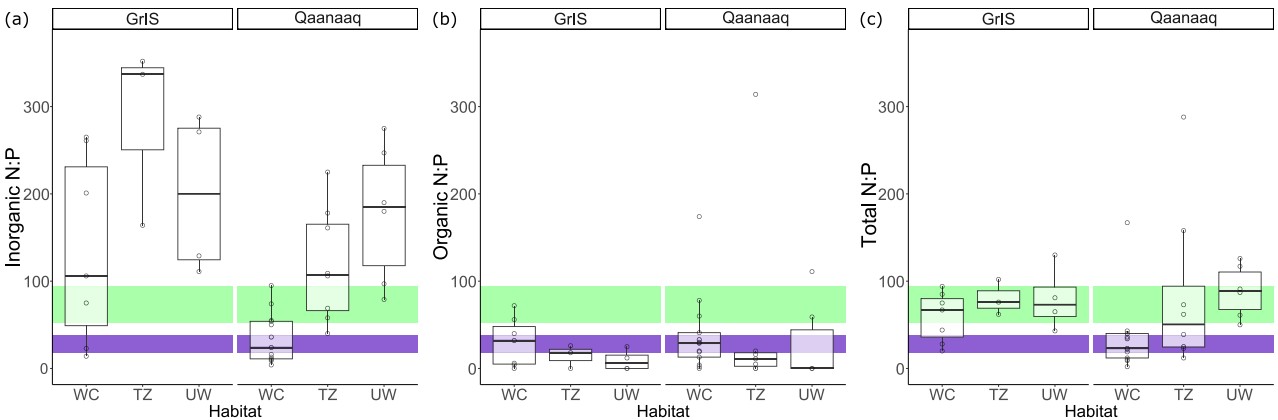

**Fig. 4 | Nitrogen to phosphorus ratios in samples from GrIS and Qaanaaq.**
**a** Boxplots of inorganic N:P ratio at different depths sampled in Qaanaaq and GrIS.
**b** Boxplots of organic N:P ratio at different depths sampled in Qaanaaq and GrIS.
**c** Boxplots of total N:P ratio at different depths sampled in Qaanaaq and GrIS.
Sample types shown are weathering crust (WC), transition zone (TZ) and

unweathered ice (UW). Boxes indicate median and interquartile ranges, while hollow points show individual measurements. The dark purple shaded area indicates the range of N:P measured in a single *Ancylonema spp.*[34]. The light green shaded area indicates the bulk cellular N:P ratio, including the standard deviation in N measurements[44].

**Table 2 | Median values and median absolute deviation (MAD) for N:P stochiometric ratios of Total, inorganic and organic fractions**

| SAMPLE | TOTAL N:P | | | | INORGANIC N:P | | | | ORGANIC N:P | | | |
|---|---|---|---|---|---|---|---|---|---|---|---|---|
| | GrIS | | Qaanaaq | | GrIS | | Qaanaaq | | GrIS | | Qaanaaq | |
| | Median | MAD | Median | MAD | Median | MAD | Median | MAD | Median | MAD | Median | MAD |
| WC | 67 | 34 | 23 | 19 | 106 | 136 | 24 | 22 | 32 | 39 | 29 | 24 |
| TZ | 76 | 21 | 51 | 39 | 284 | 22 | 108 | 76 | 18 | 12 | 14 | 16 |
| UW | 73 | 28 | 89 | 42 | 200 | 118 | 185 | 111 | 6 | 9 | 0 | 0 |

Qaanaaq, with a lower median value of 24:1 (Table 2). Organic N:P ratios followed the opposite trend, with the highest organic N:P values at the surface (Fig. 4 b). The median DON:DOP values at the surface were 32:1 and 29:1 at the GrIS site and Qaanaaq (Table 2), respectively, as opposed to the UW DON:DOP values, which were 6:1 at the GrIS site and largely below the LOD at Qaanaaq (Table 2). A combination of DIN + DON and SRP + DOP gives rise to the TDN:TDP ratios (TDN denotes total dissolved N) shown in Fig. 4c. Median TDN:TDP ratios are more similar across the three sampled depths, particularly at the GrIS site. It is noteworthy that five of the six sample groups lie within the range of bulk cellular N:P ratios.

In general, N:P ratios above the calculated bulk cellular or intracellular ratios, shown as green and purple bars in Fig. 4a, suggest that the ecosystem or the glacier ice algae, respectively, may become limited by P, whilst ratios below these values suggest a potential for N limitation. The inorganic N:P ratios of sub-surface ice (TZ and UW samples) all lie above the intracellular N:P ratios, except for some of the Qaanaaq TZ samples, which fall within this range. This suggests that ice melt due to ablation has the potential to become P-limiting at the surface, unless there are, for example, other sources of P to the surface ecosystem, particularly for *Ancylonema spp.* However, the inorganic N:P in surface ice (WC samples) at Qaanaaq lies within the range of the intracellular N:P range of *Ancylonema spp.*, suggesting P is not currently limiting for glacier ice algae at this site. Most of the GrIS surface ice (WC) samples lie above the glacier ice algae N:P ratios, suggesting that P is unlikely to limit glacier ice algal growth at present at these sites. There is evidence for balanced growth at an ecosystem level in some samples (Fig. 4a).

Previous studies have found evidence for phosphatase activity in ice surface ecosystems[43], allowing the possibility of microbial recycling of DOP. DON may be similarly recycled[31], thus the TDN:TDP of surface

and sub-surface ice may reveal whether or not the glacier ice algae and the microbial community as a whole are close to balanced growth. We note that the cellular N:P ratio of the whole ecosystem should be treated with caution since its derivation could have included dead cells (Williamson et al.[44]). Figure 4c could be regarded as remarkable, in that the majority of GrIS subsurface (TZ and UW) samples lie in the balanced growth for the microbial ecosystem, as do most of the surface (WC) ice samples. Several of the surface samples lie within the field of balanced growth for *Ancylonema spp.*, and some are intermediate between the two boxed ratios.

Qaanaaq samples show the same pattern as the inorganic N:P ratio, with ratios being lowest in the surface, and increasing with depth. Thus, if the microbiome contains functional groups able to recycle organic nutrients, there is sufficient nutrients in the melting subsurface ice to support balanced growth of the microbial community. Liebigs Law of the Minimum, from which the concept of limiting nutrients is derived, was used to describe limiting factors for growth and yield of individual crop plants in agricultural soils, and not for biologically diverse ecosystems[59]. Studies on terrestrial ecosystems have found that co-limitation of nutrients often occurs at higher latitudes, where concentrations of N and P are relatively low[60]. Surface ice ecosystems appear to be likely candidates for nutrient co-limitation if there is ready utilisation of DON and DOP (and the concentrations measured are not methodological artefacts).

**Implications for glacier ice algal growth**

Here we demonstrated that at the sites sampled on the Qaanaaq Ice cap and the GrIS in NW Greenland, both SRP and DIN are present in low, yet quantifiable concentrations within ice surfaces colonised by glacier ice algae at concentrations typical of the Dark Zone (-$10^4$–-$10^5$ cells mL$^{-1}$[19,55]). This suggests that, at the time of sampling, glacier ice

algal growth is not limited by the macronutrients, N and P. Using our measured DIN and SRP concentration in unweathered ice, along with cell concentrations, the mass of carbon in each cell, as well as stoichiometric C:N:P ratios, we estimated how much surface melt (ablation) is necessary to equal the mass of nutrient held in the glacier ice algae at each site. Our results indicate that annual ablation provides an excess of N and P relative to the glacier ice algae biomass at our locations. The N and P delivered by ablating ice is greater than the potential uptake by the algae during a single ablation season, especially considering that some glacier ice algae potentially survive from the previous winter, frozen into the surface ice[61]. This simple calculation suggests that macronutrient N and P do not limit glacier ice algal growth on seasonal timescales. We also observed higher DON and DOP concentrations at the surface, which generally exceed those of DIN and SRP. This may reflect artefacts introduced during sample processing, such as syringe or suction filtration methods, which can lyse cells, releasing intracellular DOC, DON and DOP. Cell imaging has previously shown ~10% of cells to be inactive[34], potentially due to viral[35] or fungal[36] infections. Lysis of these cells could contribute significantly to the measured DON and DOP pools, which may be recycled within the wider ecosystem. Notably, the TDN:TDP ratios we observed fall within the range expected for balanced microbial growth if the microbial consortium can access these organic nutrient forms. Finally, we used N:P ratios to visualise how short-term nutrient shortage and periodically high cell counts may be limited by DIN and/or SRP. We conclude that P may become the limiting nutrient for short periods and spatially restricted areas with high cell counts. We contend that low-level nutrient analysis is essential to understand macronutrient limitation of glacier algal growth on the Greenland Ice Sheet.

## Methods
### Site description and sampling
Ice samples were collected between 27 July and 3 August 2023 at two different locations on two different ice bodies in North West Greenland. We sampled ice on the Qaanaaq Ice Cap at an elevation between 500 and 650 m.a.s.l., on 27, 29 and 30 July (at N77° 29.8606′ W 069° 12.8812′, N77° 30.092′ W69° 09.691′ and N77° 30.150′ W69° 10.011′, respectively). The Qaanaaq Ice Cap is in the central part of the Piulip Nuna peninsula (Fig. 1b), it covers an area of 312 km² and spans an elevation range between 30–1110 m a.s.l. At the time of sampling, the ice surface was snow-free and had a shallow weathering crust (~5 cm) and dark banding from glacier ice algae and small cryoconite holes. On 3 August 2023, samples were collected from the main GrIS (at N77° 48.257′ W68° 21.044′) at an altitude of 780 m. Surface ice at this location, during sampling, had considerable biological darkening, speckled with small cryoconite holes and a shallow weathering crust (~5 cm).

Samples collected consisted of surface ice scrapings and or shallow ice cores, of which three depths were collected for analysis. Ice scrapings were collected using a field-conditioned ice axe to scrape the top layer of weathered ice in a ~2 × 2 m square and homogenized and collected into a 2 L Whirlpak® bag. These samples were categorised as weathering crust (hereafter WC) samples. Shallow cores (80–90 cm depth) were collected using a 9 cm diameter Kovax corer, targeting surface ice locations varying in surface darkness. The cores were extruded onto the inside of a clean 5.8 L whirlpak® bag, where the length of the core was quickly measured and cut into sections. From these cores, three primary 5 cm pucks were separated (Fig. 1c): (1) The top 5 cm, which contained the crumbly, active weathering crust (hereafter referred to as WC); (2) The underlying 5 cm (5–10 cm depth), which contained the transition zone from the weathering crust to the unweathered ice (hereafter referred to as TZ); 3) The last 5 cm at the base of the ice core, which consisted of fully unweathered ice (density > 0.910 g cm⁻³; hereafter UW). The height of each ice puck was measured at six points around the circumference of the puck to obtain an average height, which was used to estimate the ice density.

On the Qaanaaq cap, samples were collected from the Qaanaaq glacier, which flows southwest from the southern part of the ice cap. At this site, a total of nine cores were drilled (with WC, TZ and UW samples collected for each) and five surface ice scraping samples were collected. Additionally, three cryoconite water samples and four supraglacial stream samples were collected. At the GrIS site, four cores were drilled (and WC, TZ and UW samples collected from each) and three surface ice scrapping samples were collected. Further, one cryoconite water sample and three supraglacial stream samples were collected at this site. All samples were transported to the base site in Qaanaaq town and processed as described below.

### Sample processing
Samples were stored and partially melted in the fridge (5 °C) overnight, then stored at room temperature (~20 °C) until fully melted (approximately 2–4 hours). Samples were removed from the fridge in batches to avoid the samples warming up once melted. Immediately after melting, samples were filtered through 0.45 μm in-line PES filters, using 50 mL norm-ject syringe; previously acid-washed in 10% v/v HCl and rinsed six times with DI water. The first 5 mL of the filtered solution were discarded and approximately 45 mL of the filtered solution were aliquoted into three 15 mL falcon tubes (previously acid-washed in 10% v/v HCl and rinsed six times with DI water) and four 1.5 mL IC vials. All samples were then immediately frozen, stored at −20 °C and returned to the home laboratory on a frozen container ship. In the home laboratory, they were also stored at −20 °C until analysis.

### Nutrient analysis
Soluble reactive phosphorous (SRP), nitrite ($NO_2^-$) and total oxidised nitrogen (TON; $NO_2^- + NO_3^-$) were analysed using colorimetric methods, as outlined in Patey, et al. [29], on a custom built continuous segmented flow analyser fitted with a 250 cm Longwave Capillary Cell (LWCC, part number: LWCC-3250, WPI (Germany)) using a Tungsten Light source and a ST-VIS Miniature Spectrometer (HL-2000-LL and ST-VIS- 25, Ocean Insight (Germany)). Standard curves ranging from 5–200 nM P and 5–400 nM N were run daily on the LWCC, using TraceCERT™ phosphate standard (part num.: 38364-100 ML, Sigma-Aldrich) and Specpure© nitrite standards (part num.: 040020.AP, ThermoFisher Scientific). The LWCC calibration for TON was cross-checked using both Specpure© nitrite standards and TraceCERT™ nitrate standards (part num.: 74246-100 ML, Sigma-Aldrich). Mid-range standards consisting of 50 nM or 100 nM $PO_4^{3-}$ (for SRP and TDP, respectively) and 20 nM or 100 nM $NO_2^-$ (for $NO_2^-$ and TON, respectively) and 100 nM $NO_3^-$ (for TON) were analysed regularly throughout the run (at least every eight samples). Limits of Detection calculated as 3.3σ/S, averaged 2 nM for SRP, 1.5 nM for $NO_2^-$, and 5 nM for TON. The coefficient of variation, based on repeated analysis of 50 nM standards, was 1.6% for SRP, 1.1% for $NO_2^-$, and 5.5% for TON. All reagents were made using 18.2 MΩ cm⁻¹ water. Instrumental blanks (18.2 MΩ cm⁻¹ water) showed no signal above noise levels. The concentration of ammonium ($NH_4^+$) and concentrations above 10 ppb (~710 nM) TON, were quantified using a Gallery Aqua Master (Thermo Fisher Scientific). The default low $NH_4^+$ (LOD = 0.16 μM) and hydrazine reduction TON (LOD = 0.35 μM) Gallery Aqua Master methods were used to quantify these nutrients. Procedural blanks consisted of 18.2 MΩ cm⁻¹ water transported to the field lab in a Nalgene carboy (acid-washed with 10% v/v HCl and rinsed six times with DI water), and processed identically to samples. These blanks yielded concentrations of 2.5 nM SRP, <LOD for TON and $NO_2^-$, and 0.14 μM $NH_4^+$.

To quantify total dissolved phosphorous (TDP) and total dissolved nitrogen (TDN)– and calculate dissolved organic phosphorous (DOP) and dissolved organic nitrogen (DON)– a 7 mL aliquot of each sample was digested by adding 0.7 mL of an alkali persulphate solution and autoclaving for 30 mins, as described in Hansen and Koroleff[30]. Total nitrogen in the digested samples was then analysed using the

hydrazine reduction TON method on a Gallery Aqua Master. Total phosphorous in these extracts was analysed using the colorimetric method and custom-built analyser, as detailed above. Procedural blanks for the digestion had average concentrations of 14.2 nM TDP and 0.52 μM TDN. All digestion results were corrected using procedural blanks.

## Cell counts
Surface ice samples were counted in duplicate within 24 hours of collection, using a haemocytometer under an Olympus CX43 microscope. Magnification alternated between 100× and 200× depending on cell size. A total of 2 μL was counted per sample. Based on this volume, the detection limit for algal cells in this study was estimated at 500 cells mL$^{-1}$.

## Mineralogy and particle loading
The melted samples were filtered through 47 mm Ø, 0.22-micrometre pore size polycarbonate filters using a glass filtration unit and a vacuum pump. For each sample, the volume filtered was used to evaluate the particle loadings (mg/L of melted solution). The solids retained on the filters were used for mineralogical quantifications. To do this, first, all samples were treated with 30% $H_2O_2$ for 48 hours to remove organic matter. Following this, the mineral particles were powdered manually using an agate mortar and the mineralogy of the powders was characterized by powder X-ray diffraction (pXRD). The measurements were performed using a STOE STADI P diffractometer equipped with a Cu X-ray source, a curved Ge (111) monochromator and a MYTHEN2 detector in flat plate transmission geometry. The obtained diffraction patterns were analysed using the PROFEX software for phase identification[62].

## Statistics
Statistical analysis of the data was performed in R version 4.2.1. The Kruskal-Wallis H test was used to determine if significant differences existed for macronutrient data among ice core sections within sites. When significance was observed, Dunn's test of multiple comparisons was applied. For this study, the alpha level was set to 0.05. All errors reported in the text are one median absolute deviation about the median.

## Data availability
Additional data can be seen in the supplementary data file. The data generated in this study have been deposited in the EarthChem database under the following https://doi.org/10.60520/IEDA/114311.

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

## Acknowledgements

Special thanks for help with sample collection and processing to Katie Sipes (AU) and Christoph Keuschnig and Mamata Ghimire (GFZ). This research was funded by the European Research Council (ERC) Synergy Grant DEEP PURPLE under the European Union's Horizon 2020 Research and Innovation Programme (Grant agreement no. 856416), awarded to MT, LGB, and AA.

## Author contributions

This study was conceptualized by B.G.O. and M.T. B.G.O. wrote the original draft, with contributions from P.F. (methods and results section relating to mineralogy and particle loading). All authors reviewed and contributed to writing the manuscript. Sample collection was conducted by B.G.O., with help from Christoph Keuschnig (G.F.Z.), Katie Sipes (A.U.), T.T.J., L.G.B and A.A. Sample processing and nutrient analysis were conducted by B.G.O. P.F. and Mamata Ghimire (G.F.Z.) conducted particle loading and mineralogical analysis. Data visualisation and interpretation were conducted by B.G.O. and P.F. (related to mineralogy and particle loading results). Statistical analysis was performed by B.G.O. and T.T.J. M.T., L.G.B., and A.A. obtained the funding for the Deep Purple ERC Synergy project.

## Competing interests

The authors declare no competing interests.
