## [Peer Review file · Nature Communications]

Ablation provides key macronutrients (nitrogen and phosphorous) to glacier ice algae in NW Greenland.

Corresponding Author: Dr Beatriz Gill Olivas

Version 0:

Reviewer comments:

Reviewer #1

(Remarks to the Author)

This is a well written and presented paper on the study of N and P dynamics within the upper icecap zones, and whether this may explain microbial growth distribution and dynamics. I have very few comments as all the information is mainly well presented and carried out correctly, the key item is to double check the values in the text and the figures/tables to make sure they match to the correct values.

Introduction:

Line 19: "Intro" should be in full "introduction"

L23: change to "The growth of these algae decreases..." as plural

L31: delete "On the other hand," or replace with However...

Results:

Line 89: "2.8 to 3.8% of the cells counted at Qaanaaq (mean value: 2.4×10^5 cells mL⁻¹) and GrIS sites (mean: 0.75×10^5 cells mL⁻¹), respectively" – these values do not match what you are reporting in the supplementary data where it says 2.4% and 0.75×10^5 – please check all values for consistency and accuracy to your original dataset as I do not know which values are the correct/true ones

Line 92: first sentence too long and needs rewording as I am not sure if you are presenting your findings here or values of NEGIS or NEEM

Line 92: also for this paragraph please report in the same order of which the Table 1 is presenting data (so Total P, then SRP then DOP).

Line 92: for this paragraph please refer to where your data are in tables and figs

Line 101: you need to say what your range is if comparing to another range

Table 1,2,3: these can all be in the same table

Fig 2: add info: "weathering crust (WC), transition zone (TZ) and unweathered ice (UW)" to help the reader

Fig 4 text: Nordenskiöldii should be lowercase n

L183-184: the cell numbers do not match table S1

L179: I suggest all of this long calculation-text be summarised as an equation somewhere in the document to make it easier to follow

Suggest addition of findings from this recent paper to put values into context too (note I am not an author): Micromelt sampling of the glacier algal nutrient environment. Madeleine Lewis et al. FEMS Microbiology Ecology, Volume 101, Issue 11, November 2025, fiaf098, <https://doi.org/10.1093/femsec/fiaf098>

Reviewer #2

(Remarks to the Author)

This study focuses on the nitrogen (N) and phosphorus (P) macronutrient limitation of glacial ice algal growth in northwestern Greenland. By employing a custom high-sensitivity analytical instrument to achieve nanomolar-level nutrient concentration detection, combined with mass balance calculations and N:P ratio analysis, it draws the core conclusion that "N and P do not limit glacial ice algal growth on a seasonal scale." The research topic holds significant implications for glacial ecology and climate interactions, with a logically sound experimental design and relatively solid data support. However, there is room for improvement in research depth, methodological details, conclusion generalizability, and certain mechanistic discussions.

Specific Revision Comments

Research Content and Mechanistic Discussion

1. The study infers that the higher soluble reactive phosphorus (SRP) concentration in the surface layer than in the deep layer is associated with surface dust dissolution and mineralization, and clarifies the differences in dust mineral composition between the Qaanaaq Ice Cap and Greenland Ice Sheet (GrIS) sampling sites. However, it fails to quantify the specific contribution ratio of different minerals to N and P supply. It would enhance the persuasiveness of mechanistic explanations if laboratory dust dissolution simulation experiments were supplemented, controlling key environmental factors such as temperature and pH (matching glacial meltwater characteristics), measuring the N and P release rates of different mineral phases, and clarifying the influence weight of geological background differences on nutrient supply.
2. Regarding the phenomenon that the nitrite (NO_2^-) concentration in the surface layer is significantly higher than that in the deep layer, only atmospheric deposition or nitrification is inferred as the source without distinguishing their contribution ratios. It would improve the research on nitrogen cycling mechanisms if isotope tracing technology (e.g., ^{15}N labeling) or functional gene detection (e.g., nitrification-related *amoA* gene) were integrated to accurately identify the main source of NO_2^- , while supplementing data on cryoconite granule distribution density and analyzing its correlation with nitrification intensity.
3. The study mentions that algal cell lysis during sample processing may overestimate dissolved organic nitrogen (DON) and dissolved organic phosphorus (DOP) concentrations. Supplementary Table S1 and the associated calculations detail the estimation of DON/DOP derived from cell lysis. However, the study still lacks verification of the natural cell lysis rate under in situ conditions. It would facilitate the revision of relevant conclusions if a fluorescence staining method (e.g., SYTOX Green) were used to determine the mortality rate of glacial ice algae under natural conditions, quantify the DON and DOP produced by natural lysis, and evaluate its impact on the measured data.

Research Methods and Data Analysis

1. The mass balance calculation for unweathered ice melt equivalent provides steps for estimating N/P storage in algae and required ablation, but it still assumes 100% absorption efficiency of dissolved inorganic nitrogen (DIN) and SRP by algae, without considering nutrient loss through supraglacial runoff and competitive consumption by other microorganisms. It would improve the reliability of conclusions if laboratory culture experiments were conducted to determine the actual N and P absorption efficiency of algae, nutrient loss coefficients were calculated based on supraglacial runoff monitoring data, microbial competitive consumption parameters were introduced, and the existing balance model was revised.
2. A custom continuous segmented flow analyzer is used for low-concentration nutrient detection, but its results have not been compared and verified with mainstream high-resolution mass spectrometry (e.g., LC-MS/MS). It would enrich the nutrient composition analysis if representative samples (covering different depths and nutrient concentration ranges) were selected for parallel detection by both methods to verify the accuracy of the custom instrument's results; meanwhile, attempts could be made to identify more low-concentration N and P forms through mass spectrometry.
3. The N:P ratio analysis is based on the median value of the overall sample, without considering the spatial heterogeneity of intracellular N:P ratios in algae. It would avoid misjudgment of limitation potential caused by overall mean analysis if single-cell level N:P ratio analysis (e.g., micro spectroscopic technology) was supplemented to explore the nutritional status differences of algal cells in different microhabitats at the same sampling site.

Conclusion Generalizability and Extended Research

1. This study only sampled two sites in northwestern Greenland in July-August 2023 (mid-ablation season), without covering different ablation stages and other regions (e.g., the dark zone in southeastern Greenland). It would enhance the comprehensiveness of the research if the temporal and spatial limitations of the conclusions were clearly stated in the discussion section, prospects for multi-season and multi-region studies were supplemented, or preliminary analysis of the regional specificity of nutrient limitation was conducted by comparing existing data from different regions.
2. The study mentions the positive feedback of "melting - nutrient release - algal growth - further melting" but does not quantify the contribution ratio of N and P supply in this feedback. It would strengthen the research depth and application value if a coupled model of glacial ablation and algal growth incorporating N and P cycling were constructed by combining numerical simulation and field observation data, and the intensity of nutrient supply's role in the positive feedback was

quantified.

3. The potential impact of the continuous increase in glacial ablation rate under global warming on the balance between N and P supply and algal demand is not discussed. It would expand the research prospect if the CMIP6 climate model prediction results were integrated in the discussion to simulate changes in nutrient release and algal growth rate under different future warming scenarios, and analyze whether the "nutrient surplus" state would be disrupted or new limiting factors would emerge.

Experimental Details and Data Presentation

1. The study mentions "discarding the first 5 mL of filtered solution after filtration" but does not explain the specific basis for this operation (e.g., whether pre-experiments were conducted to verify pollutant residue in the first 5 mL of solution). It would improve the rigor of the experimental method if relevant verification data or reference support were supplemented to clarify the rationality of this operation.

2. Supplementary Table S1 provides key data such as cell counts, C/N/P mass estimates, and required ablation, but it would be more user-friendly if additional annotations were added to clarify the derivation of parameters (e.g., citation of sources for the 8.8 pM C cell⁻¹ value and C:N:C:P ratios) and unit consistency checks (e.g., confirmation of pmol/ml vs. other nutrient concentration units).

3. The calculation process for cell lysis estimates and unweathered ice melt equivalent is detailed, but it would enhance reproducibility if the raw data used in the calculations (e.g., raw cell count data, individual sample DIN/SRP/TDN/TDP concentrations) were included in the supplementary materials, allowing readers to replicate the calculations independently. This study addresses the challenge of accurately determining low-concentration nutrients using traditional methods through innovative detection technology, and its core conclusions have certain scientific value. It is recommended that the authors supplement relevant experimental data, revise the analysis model, improve mechanistic discussions and data presentation in response to the above comments, so as to further enhance the rigor, depth, and generalizability of the research.

Response to Reviewer 1:

We would like to thank reviewer 1 for their commentary on our manuscript and for engaging with the subject. Please see below for our point-by-point response to each of the comments raised (the response by the authors is in bold and italicized sans serif font):

--

Reviewer #1 (Remarks to the Author):

This is a well written and presented paper on the study of N and P dynamics within the upper icecap zones, and whether this may explain microbial growth distribution and dynamics. I have very few comments as all the information is mainly well presented and carried out correctly, the key item is to double check the values in the text and the figures/tables to make sure they match to the correct values.

Introduction:

Line 19: “Intro” should be in full “introduction” - ***Done***

L23: change to “The growth of these algae decreases...” as plural - ***Done***

L31: delete “On the other hand,” or replace with However... - ***Replaced to “However”***

Results:

Line 89: “2.8 to 3.8% of the cells counted at Qaanaaq (mean value: 2.4×10^5 cells mL⁻¹) and GrIS sites (mean: 0.75×10^5 cells mL⁻¹), respectively” – these values do not match what you are reporting in the supplementary data where it says 2.4% and 0.75×10^5 – please check all values for consistency and accuracy to your original dataset as I do not know which values are the correct/true ones

We reviewed the values, which were correct. However, the Supplementary reports 7.5×10^4 , which has been changed to 0.75×10^5 to maintain consistency.

Line 92: first sentence too long and needs rewording as I am not sure if you are presenting your findings here or values of NEGIS or NEEM.

We have broken the sentence into two and added some further clarification regarding the values. It now reads as:

“The SRP concentrations reported here (Table 1) are an order of magnitude higher than those measured in firn cores from the North East Greenland Ice Stream (NEGIS) ice core sampling site, within the GrIS accumulation zone (mean SRP at NEGIS: 2.7 nM^{21}). The concentrations of SRP measured in our samples were mostly within the range measured in the NEEM ice core samples (SRP within NEEM: $3.0 - 32 \text{ nM}^{20}$). However, some of the SRP concentrations in our samples were up to four times higher than the upper values found at NEEM (Table 1).”

Line 92: also for this paragraph please report in the same order of which the Table 1 is presenting data (so Total P, then SRP then DOP).

We have rearranged the paragraph so TDP is presented before SRP.

Line 92: for this paragraph please refer to where your data are in tables and figs

We have added more mentions to table 1 for clarity.

Line 101: you need to say what your range is if comparing to another range

We have added “(means ranging from 36 – 142 nM; Table 1)” for clarity.

Table 1,2,3: these can all be in the same table

We have combined them as suggested.

Fig 2: add info: “weathering crust (WC), transition zone (TZ) and unweathered ice (UW)” to help the reader

We have added a line to the caption that reads “Sample types shown are weathering crust (WC), transition zone (TZ) and unweathered ice (UW)”. We have added this to all the figure captions for clarity.

Fig 4 text: Nordenskiöldii should be lowercase n - **Done**

L183-184: the cell numbers do not match table S1

Values had been rounded to 1 s.f., we have changed it to 2 s.f. for consistency.

L179: I suggest all of this long calculation-text be summarised as an equation somewhere in the document to make it easier to follow

We have added four equations to the supplementary to make it easier for the reader.

Suggest addition of findings from this recent paper to put values into context too (note I am not an author): Micromelt sampling of the glacier algal nutrient environment. Madeleine Lewis et al. FEMS Microbiology Ecology, Volume 101, Issue 11, November 2025, fiaf098, <https://doi.org/10.1093/femsec/fiaf098>

We have included a mention to it in the introduction, to acknowledge that micromelt in surface ice might have higher concentrations of nutrients. It also highlights the difficulty of quantifying these concentrations. Most of the PO_4^{3-} , NO_3^- and NO_2^- values reported in this paper were below their reported LOD (limit of detection). This may be an artefact of blank corrections, however I could not verify this from the paper and there was no data repository link to further assess the data. Therefore, we have included the following sentence to our manuscript “Recent studies suggest that concentrations of various nutrients may be higher in the “micromelt” (i.e. a thin layer of melt water of the surface of melting ice crystals) than in the bulk ice crystals²⁴. However, even in these samples the concentration of PO_4^{3-} , NO_3^- and NO_2^- was largely below the limit of detection (LOD)²⁴” (line 44 - 47).

We note that these micromelt concentrations do not impact on the mass balance of nutrients in the surface ice, since the remaining parent ice is nutrient depleted.

Response to Reviewer 2:

We would like to thank reviewer 2 for their commentary on our manuscript. The review raises some interesting questions. However, we feel that much of the suggested additional work is not only beyond the scope of **this** study, but beyond the current leading edge of the science. We hope that others will also be inspired to pose similar questions and therefore continue to move the field forward. Our point-by-point response can be seen in bold and italicized sans-serif font.

--

Reviewer #2 (Remarks to the Author):

This study focuses on the nitrogen (N) and phosphorus (P) macronutrient limitation of glacial ice algal growth in northwestern Greenland. By employing a custom high-sensitivity analytical instrument to achieve nanomolar-level nutrient concentration detection, combined with mass balance calculations and N:P ratio analysis, it draws the core conclusion that "N and P do not limit glacial ice algal growth on a seasonal scale." The research topic holds significant implications for glacial ecology and climate interactions, with a logically sound experimental design and relatively solid data support. However, there is room for improvement in research depth, methodological details, conclusion generalizability, and certain mechanistic discussions.

Specific Revision Comments

Research Content and Mechanistic Discussion

1. The study infers that the higher soluble reactive phosphorus (SRP) concentration in the surface layer than in the deep layer is associated with surface dust dissolution and mineralization, and clarifies the differences in dust mineral composition between the Qaanaaq Ice Cap and Greenland Ice Sheet (GrIS) sampling sites. However, it fails to quantify the specific contribution ratio of different minerals to N and P supply. It would enhance the persuasiveness of mechanistic explanations if laboratory dust dissolution simulation experiments were supplemented, controlling key environmental factors such as temperature and pH (matching glacial meltwater characteristics), measuring the N and P release rates of different mineral phases, and clarifying the influence weight of geological background differences on nutrient supply.

We are grateful that the reviewer has engaged with the text enough to make these observations. However, the source of the nutrients is not the current question we are aiming to answer in this study. Our study aims to quantify the current levels of N and P at the ice surface and highlight the importance of using instruments which are able to quantify these low-level nutrients, since the impact of even low levels of nutrients has such an impact on the numbers of glacier ice algal cells that surface ice at these locations can maintain. Understanding the relative contributions of nutrients from different dust sources is a great future research project that this paper has justified the need for.

2. Regarding the phenomenon that the nitrite (NO_2^-) concentration in the surface layer is significantly higher than that in the deep layer, only atmospheric deposition or nitrification is inferred as the source without distinguishing their contribution ratios. It would improve the research on nitrogen cycling mechanisms if isotope tracing technology (e.g., ^{15}N labeling) or functional gene detection (e.g., nitrification-related amoA gene) were integrated to accurately identify the main source of NO_2^- , while supplementing data on cryoconite granule distribution density and analyzing its correlation with nitrification intensity.

We highlight the increase in NO₂⁻ at the surface relative to deeper within the ice, as an interesting result. However, we are careful to not overinterpret these results, particularly as it is only a small fraction of the inorganic N pool. The time and associated cost necessary to conduct this additional work makes it not only impractical but unreasonable given the scope of the paper. We feel that this is another great future research project in the making.

3. The study mentions that algal cell lysis during sample processing may overestimate dissolved organic nitrogen (DON) and dissolved organic phosphorus (DOP) concentrations. Supplementary Table S1 and the associated calculations detail the estimation of DON/DOP derived from cell lysis. However, the study still lacks verification of the natural cell lysis rate under in situ conditions. It would facilitate the revision of relevant conclusions if a fluorescence staining method (e.g., SYTOX Green) were used to determine the mortality rate of glacial ice algae under natural conditions, quantify the DON and DOP produced by natural lysis, and evaluate its impact on the measured data.

There is, to date, no information relating to the lysis and mortality rate of these algae, it is an open question at the leading edge of the science. We estimated the cell lysis that would account for the DON and DOP concentrations measured to put our results into perspective and to avoid over-interpretation of the organic data. We have included the current best estimates of dead or “inactive” cells which have been measured (~10%; in line 142-145), to acknowledge that algal cell lysis may also occur naturally, even if our current understanding of the mortality rates of these algae is limited. The very act of filtration almost certainly lyses cells, but the experimentation required to examine this effect is non-trivial because of the environmental stresses experienced by the glacier ice algae during their collection and in preparation for their experimentation and subsequent counting.

Research Methods and Data Analysis

1. The mass balance calculation for unweathered ice melt equivalent provides steps for estimating N/P storage in algae and required ablation, but it still assumes 100% absorption efficiency of dissolved inorganic nitrogen (DIN) and SRP by algae, without considering nutrient loss through supraglacial runoff and competitive consumption by other microorganisms. It would improve the reliability of conclusions if laboratory culture experiments were conducted to determine the actual N and P absorption efficiency of algae, nutrient loss coefficients were calculated based on supraglacial runoff monitoring data, microbial competitive consumption parameters were introduced, and the existing balance model was revised.

We acknowledge that this mass balance calculation is not perfect, this is acknowledged within the text:

“in reality the amount of ablation required will be greater because the DIN and SRP uptake efficiency of the algae is unlikely to be 100%, and there is a nutrient demand from other microbes in the microbial ecosystem” (line 202-204) and again:

“This too assumes that the glacier ice algae can directly take up DOP and DON with an efficiency of 100% and/or that heterotrophic microorganisms take up and recycle DOP and DON to inorganic forms with 100% efficiency (i.e., without no loss to downstream ecosystems)” (line 213-215).

These calculations also assume that the bloom develops completely within one melt season, which is unlikely, as we also mention in our manuscript:

“The N and P delivered by ablating ice is greater than the potential uptake by the algae during a single ablation season, especially considering that some glacier ice algae potentially survive from the previous winter, frozen into the surface ice⁶⁰”.

The laboratory culturing experiments are non-trivial and are beyond the current leading edge of the field. Only within the last few years have research groups been able to successfully culture them in the lab, as outlined by Jensen et al. (2023) paper “The dark art of cultivating glacier ice algae”. This, together with the other suggestions already made by the reviewer takes us well beyond what is feasible to report in one short journal paper, and would require several grants of multiple years to achieve.

2. A custom continuous segmented flow analyzer is used for low-concentration nutrient detection, but its results have not been compared and verified with mainstream high-resolution mass spectrometry (e.g., LC-MS/MS). It would enrich the nutrient composition analysis if representative samples (covering different depths and nutrient concentration ranges) were selected for parallel detection by both methods to verify the accuracy of the custom instrument's results; meanwhile, attempts could be made to identify more low-concentration N and P forms through mass spectrometry.

The use of colorimetric/spectrophotometric methods to quantify N and P in water is standard in many fields, particularly for environmental chemistry. Although our set-up and the use of a LWCC has not been use in supraglacial samples and is not widely used due to the difficulties associated with setting up the method and associated maintenance, it has previously been used in marine sciences. We reference Patey et al. (2008) as our main source in the methods as we replicate their methods, and earlier studies by Zhang et al. (2002) and Zhang et al. (2003) have used variations of this method. We use certified standards and standard curves to calibrate our method, and have included some more details to the methods section to clarify this point: “Standard curves ranging from 5 – 200 nM P and 5 – 400 nM N were run daily on the LWCC, using TraceCERT™ phosphate standard (part num.: 38364-100ML, Sigma-Aldrich) and Specpure© nitrite standards (part num.: 040020.AP, ThermoFisher Scientific). The LWCC calibration for TON was cross-checked using both Specpure© nitrite standards and TraceCERT™ nitrate standards (part num.: 74246-100ML, Sigma-Aldrich). Mid-range standards consisting of 50 nM or 100 nM PO₄³⁻ (for SRP and TDP, respectively) and 20 nM or 100nM NO₂⁻ (for NO₂⁻ and TON, respectively) and 100 nM NO₃⁻ (for TON) were analysed regularly throughout the run (at least every eight samples).” (line 367-374).

Finally, although LC-MS/MS would add a different dimension to our understanding of the composition of the organic fractions, it is no way a “mainstream” method of doing this, neither would it offer a direct comparison to the data presented here.

3. The N:P ratio analysis is based on the median value of the overall sample, without considering the spatial heterogeneity of intracellular N:P ratios in algae. It would avoid misjudgment of limitation potential caused by overall mean analysis if single-cell level N:P ratio analysis (e.g., micro spectroscopic technology) was supplemented to explore the nutritional status differences of algal cells in different microhabitats at the same sampling site.

The intra-cellular N:P ratios discussed and their variability were part of a study published earlier in the year by Halbach et al. (2025). We address the variability that can be found within the population by specifically highlighting the wide range of N:P values that were measured in this study and including that variability in our plots (i.e. using a shaded area in Fig. 4 rather than a simple line). It is beyond the scope of this

study to conduct this same micro spectroscopic analysis on cells found at these sites. Quantifying the N:P range in different microhabitats is a significant and costly research task requiring a separate study of at least a year, which would be justified from the findings of this manuscript, but not necessary for the interpretation of our results. Put simply, this is a foundational study that underscores the need for future funding and continued advancement in the field.

Conclusion Generalizability and Extended Research

1. This study only sampled two sites in northwestern Greenland in July-August 2023 (mid-ablation season), without covering different ablation stages and other regions (e.g., the dark zone in southeastern Greenland). It would enhance the comprehensiveness of the research if the temporal and spatial limitations of the conclusions were clearly stated in the discussion section, prospects for multi-season and multi-region studies were supplemented, or preliminary analysis of the regional specificity of nutrient limitation was conducted by comparing existing data from different regions.

As we highlight throughout the manuscript, there are very few studies that have quantified supraglacial nutrient concentrations. Even the studies that are available have limitations due to the concentrations being at or below the limit of detection. We have included some of the studies which look at nutrients in other polar regions (e.g. Svalbard and the alps) for context in the introduction. We have referenced the few results available looking at nutrients at the surface of the GrIS to give context to our results both geographically and the potential differences that may arise with within different times in the melt-season. Particularly, we make reference to studies from Kjæer et al. (2013 and 2015; which looked at P within ice and firn cores taken in the accumulation zone of the GrIS – incidentally using the same LWCC techniques employed in this paper), Holland et al. (2019 and 2022; which attempted to quantify P and N during the early melt season in the dark zone of the GrIS with limited success due to a high proportion of their values being below the LOD) and Telling et al. (2012; which analysed N in ice samples from Southern Greenland further into the melt season) and to a lesser extent Stibal et al. (2011; also measured nutrients across a transect in Southern Greenland but results were often below the LOD). Currently, our understanding of the variability of these nutrients across regions and seasons is beyond what can be achieved in a single study, and requires further dedicated research funding of several years' duration.

2. The study mentions the positive feedback of "melting - nutrient release - algal growth - further melting" but does not quantify the contribution ratio of N and P supply in this feedback. It would strengthen the research depth and application value if a coupled model of glacial ablation and algal growth incorporating N and P cycling were constructed by combining numerical simulation and field observation data, and the intensity of nutrient supply's role in the positive feedback was quantified.

This is a grand design well beyond the scope of this short journal paper, which would be a significant foundational block to win the research funding and time needed to undertake such a project.

3. The potential impact of the continuous increase in glacial ablation rate under global warming on the balance between N and P supply and algal demand is not discussed. It would expand the research prospect if the CMIP6 climate model prediction results were integrated in the discussion to simulate changes in nutrient release and algal growth rate under different future warming scenarios, and analyze whether the "nutrient surplus" state would be disrupted or new limiting factors would emerge.

We do not yet know how these climate predictions are likely to affect the growth of glacier ice algae (or the wider glacial microbiome), therefore it would be highly

speculative to try to predict how these changes may affect the nutrient demands. There are too many uncertainties at this time to be able to speculate how future warming may affect the growth of glacier ice algae and how their nutrient demands may be affected. This is another grand design well beyond the scope of this short journal paper, which would be a significant foundational block to win the research funding and time needed to undertake such a project.

Experimental Details and Data Presentation

1. The study mentions "discarding the first 5 mL of filtered solution after filtration" but does not explain the specific basis for this operation (e.g., whether pre-experiments were conducted to verify pollutant residue in the first 5 mL of solution). It would improve the rigor of the experimental method if relevant verification data or reference support were supplemented to clarify the rationality of this operation.

Discarding some of the filtrate (first 0.5 -2 ml) is a relatively standard procedure to precondition filters. This is done to remove contaminants or extractables and to prevent/reduce adsorption bias. It often appears as a recommendation on user manuals for syringe filters (e.g. "User Guide Nonsterile 33 mm Millex® Syringe Filters Millex®-LG, LCR, GV, HV, GN, HN, GP, HP, FG, FH" by Millipore, recommends "you discard the first 1 mL"). We chose to increase the volume to 5 mL as a preventative measure to reduce the chances of interference from extractables or from adsorption bias. Since it is a standard procedure, we did not do additional testing, however, we analysed procedural blanks that suggest there is no significant contamination arising from the filtering procedure.

2. Supplementary Table S1 provides key data such as cell counts, C/N/P mass estimates, and required ablation, but it would be more user-friendly if additional annotations were added to clarify the derivation of parameters (e.g., citation of sources for the 8.8 pM C cell⁻¹ value and C:N/C:P ratios) and unit consistency checks (e.g., confirmation of pmol/ml vs. other nutrient concentration units).

The citations for these values are all included in the text describing the calculations. The 8.8 pM C cell⁻¹ value is a conversion from pg C/cell to pmol C/cell.

3. The calculation process for cell lysis estimates and unweathered ice melt equivalent is detailed, but it would enhance reproducibility if the raw data used in the calculations (e.g., raw cell count data, individual sample DIN/SRP/TDN/TDP concentrations) were included in the supplementary materials, allowing readers to replicate the calculations independently.

The data will be uploaded onto a data repository to facilitate this.

This study addresses the challenge of accurately determining low-concentration nutrients using traditional methods through innovative detection technology, and its core conclusions have certain scientific value. It is recommended that the authors supplement relevant experimental data, revise the analysis model, improve mechanistic discussions and data presentation in response to the above comments, so as to further enhance the rigor, depth, and generalizability of the research.